# Histological and Immunohistochemical Studies to Determine the Mechanism of Cleft Palate Induction after Palatal Fusion in Mice Exposed to TCDD

**DOI:** 10.3390/ijms23042069

**Published:** 2022-02-13

**Authors:** Chisato Sakuma, Hideto Imura, Tomohiro Yamada, Azumi Hirata, Yayoi Ikeda, Masaaki Ito, Nagato Natsume

**Affiliations:** 1Division of Research and Treatment for Oral and Maxillofacial Congenital Anomalies, School of Dentistry, Aichi-Gakuin University, 2-11 Suemori-Dori, Chikusa-ku, Nagoya 464-8651, Japan; char153690141@gmail.com (C.S.); masa1119mile@gmail.com (M.I.); natsume@dpc.agu.ac.jp (N.N.); 2Section of Oral and Maxillofacial Surgery, Division of Maxillofacial Diagnostic and Surgical Science, Faculty of Dental Science, Kyushu University, 3-1-1 Maidashi, Higashi-ku, Fukuoka 812-8582, Japan; yamada.tomohiro.733@m.kyushu-u.ac.jp; 3Department of Anatomy and Cell Biology, Faculty of Medicine, Osaka Medical College, 2-7 Daigaku-Machi, Takatsuki 569-8686, Japan; an1026@osaka-med.ac.jp; 4Department of Anatomy, School of Dentistry, Aichi-Gakuin University, 1-100 Kusumoto-cho, Chikusa-ku, Nagoya 464-8650, Japan; yayoi@dpc.agu.ac.jp

**Keywords:** cleft lip, cleft palate, palatal fusion, rupture, TCDD, epithelial cell adhesion, cell proliferation

## Abstract

Rupture of the basement membrane in fused palate tissue can cause the palate to separate after fusion in mice, leading to the development of cleft palate. Here, we further elucidate the mechanism of palatal separation after palatal fusion in 8–10-week-old ICR female mice. On day 12 of gestation, 40 μg/kg of 2,3,7,8-Tetrachlorodibenzo-p-dioxin (TCDD), sufficient to cause cleft palate in 100% of mice, was dissolved in 0.4 mL of olive oil containing toluene and administered as a single dose via a gastric tube. Fetal palatine frontal sections were observed by H&E staining, and epithelial cell adhesion factors, apoptosis, and cell proliferation were observed from the anterior to posterior palate. TUNEL-positive cells and Ki67-positive cells were observed around the posterior palatal dissection area of the TCDD-treated group. Moreover, in fetal mice exposed to TCDD, some fetuses exhibited cleft palate dehiscence during fusion. The results suggest that palatal dehiscence may be caused by abnormal cell proliferation in epithelial tissues, decreased intercellular adhesion, and inhibition of mesenchymal cell proliferation. By elucidating the mechanism of cleavage after palatal fusion, this research can contribute to establishing methods for the prevention of cleft palate development.

## 1. Introduction

Cleft lips and palates are the most common external deformities among congenital anomalies in humans. The incidence of cleft lip and cleft palate is highest among Asians, with an incidence of 1 in 500–600 in the Japanese population [1,2]. In a previous study analyzing births in the Tokai region of Japan over 37 years, the incidence of cleft lip and palate ranged from 0.11% to 0.21% [3]. Among the two deformities, the estimated incidence of cleft palate is approximately 33% [4]. However, the underlying mechanism of cleft palate induction has not been elucidated.

In the human palate, the palatine process [5], which is the precursor to palate formation, occurs on both sides of the tongue at approximately 6 weeks of fetal age. At approximately 8 weeks of fetal age, the tongue moves downward, and the left and right palatine processes are elevated to a horizontal position above the tongue. The horizontal left and right palatine processes then elongate toward the midline and come into contact with each other at approximately 10 weeks of fetal age. At the midline contact point, the palatine processes on both sides form an epithelial cord and begin to fuse together. At approximately 12 weeks of fetal age, the epithelial cords disappear because of apoptosis of the epithelial cells, completing palatal fusion and forming the palate. Cleft palate is thought to be an abnormality caused during palate development via one of the following mechanisms the left and right palatine processes are not elevated, and horizontal dislocation does not occur [6]; palatine processes are elevated but do not grow horizontally and do not make contact [7]; or left and right palatine processes make contact and do not fuse because of horizontal dislocation and horizontal growth of the palatine processes [8,9] and cleft palate, resulting in separation of the palate after fusion of the palatine processes [10,11]. Genetic factors include such as mutations in TGFβ, MSX1, TBX22, IRF6, and MEOX2, and Hoxc, HOXB3, and TFAP2A [12,13,14,15,16,17]. The multifactorial threshold theory suggests that environmental factors, such as alcohol and tobacco consumption, maternal obesity, and exposure to chemicals such as TCDD and vitamin A [18,19,20] interact to cause cleft palate once a certain threshold is exceeded. TCDD is considered the most toxic of all dioxins and causes hydronephrosis and cleft palate in mice by exerting teratogenic effects [21].

In previous laboratory experiments, we also constructed mouse models of cleft palate formation. Yamada et al. [22] reported that 100% of pregnant mice administered a critical concentration of TCDD during fetal palate formation developed cleft palate. Furthermore, Imura et al. [23] reported that when fetuses were removed at 18 days of fetal life from pregnant mice administered a critical concentration of TCDD, 100% of the fetuses developed cleft palate; however, when fetuses were removed at 14, 15, and 16 days of fetal life, only 4%, 17%, and 13% of the fetuses developed cleft palate, respectively, indicating that cleft palate formation occurred after palatal fusion. Furthermore, we previously showed that rupture in the basement membrane of fused palate tissue can cause the palate to separate after fusion [24].

Therefore, in this study, we elucidate the mechanism of cleft palate development after palatal fusion in mice.

## 2. Results

### 2.1. Histological Observations

Hematoxylin and eosin (H&E) staining in the control group showed that the palate fused from the anterior to the posterior end, and epithelial cords were observed in the midline of the palatal shelf (Figure 1A, B, arrows). In the TCDD-treated group, anterior palate fusion and posterior palate dehiscence were histologically confirmed in 15 of 38 fetuses exhibiting anterior palate fusion and posterior palate dehiscence under stereomicroscopy; representative H&E staining images are shown in Figure 1C, D. In the control group, cells were found in the entire palatal shelf at anterior palatal fusion (Figure 1A), whereas cell density in the nasolacrimal mucosa was sparse in the TCDD-treated group (Figure 1C, arrow). In addition, separation of the palate from the oral cavity side was observed in the posterior part of the palate (Figure 1D, arrow).

### 2.2. Observation of Epithelial and Interepithelial Cell Adhesion Factors and Basement Membrane

In the anterior and posterior palates of the control group, E-cadherin, an epithelial cell adhesion factor, was positive for epithelial cells in the nasal and oral mucosa and epithelial cords, and laminin, a component of the basement membrane, showed continuous staining images (Figure 2A–C,G–I). In the anterior palatine fossa of the TCDD-treated group, E-cadherin was positive for epithelial cells in the nasal and oral mucosa and epithelial cords, and laminin was stained continuously in the nasal mucosa and epithelial cords (Figure 2D–F), but showed discontinuous staining in the oral mucosa (Figure 2E,F, arrowheads). In the posterior part of the palatal dehiscence, the oral mucosa showed dehiscence (Figure 2J–L, small arrows), laminin showed discontinuous staining in the oral mucosa (Figure 2K,L, arrowheads), and epithelial cells in the same area were negative for E-cadherin (Figure 2J–L, arrowheads). In addition, E-cadherin was positive for epithelial cells in the epithelial cord near the palatine detachment (Figure 2J–L, large arrow), and laminin staining was observed (Figure 2K,L, large arrow).

We also observed and examined localization of β-catenin and α-catenin, which are epithelial cell adhesion factors that form a complex with E-cadherin. In the control group, epithelial cells were positive for β-catenin and α-catenin in the nasal and oral mucosa and epithelial cords in the anterior and posterior palates (Figure 3A,B,E,F). In the TCDD-treated group, β-catenin and α-catenin were positive for epithelial cells in the nasal and oral mucosa in the anterior palatal fusion area, as in the control group, but negative in the median nasal mucosa (Figure 3C,D). Epithelial cells were positive for β-catenin and α-catenin in the epithelial cords of the midline palate (Figure 3C,D, arrowheads). In the posterior palatine detachment, epithelial cells were positive for β-catenin and α-catenin in the nasal and oral mucosa (Figure 3G,H), and β-catenin and α-catenin were positive for epithelial cells in the epithelial cord near the palatine detachment (Figure 3G,H, arrowheads).

### 2.3. Observation of Apoptosis

In the control group, TUNEL-positive cells were observed in the anterior and posterior palates, indicating that apoptosis occurred in the epithelial cords (Figure 4A,B, arrowheads). In the TCDD-treated group, TUNEL-positive cells were found in the epithelial cords in the anterior palatal fusion area as well as in the control group (Figure 4C, arrowhead). In the posterior palatal dissection area, TUNEL-positive cells were found around the dissection area (Figure 4D, arrowhead).

### 2.4. Observation of Cell Proliferation

In the control group, cells staining positive for Ki67, a cell proliferation marker, were observed in the nasal and oral mucosa and around the epithelial cord in the anterior palate (Figure 5A, arrowhead). In the posterior palate, Ki67-positive cells were found in the nasal and oral mucosa and the entire palatal shelf (Figure 5B, arrowhead). In the TCDD-treated group, Ki67-positive cells were found in the anterior palatal fusion area on the nasal and oral sides of the mucosa and around the epithelial cord (Figure 5C, arrowhead). In the posterior part of the palatal dissection, Ki67-positive cells were found around the dissection area (Figure 5D, arrowhead).

## 3. Discussion

TCDD, an endocrine disruptor, is known to induce a variety of biological toxicities and exhibits acute toxicity, immunotoxicity, reproductive toxicity, developmental toxicity, and carcinogenicity [25]. In particular, disorders related to tissue developmental toxicity such as hydronephrosis and cleft palate are caused by TCDD exposure during fetal life [21,26], as well as disorders related to cellular developmental toxicity such as epithelial dysplasia [27]. Moreover, genetic mutations and genetic polymorphisms in several genes, including extracellular matrix genes, soluble factors, and enzymes responsible for extracellular matrix remodeling, may play a role in the etiology of cleft palate [28]. In addition, maternally expressed gene 3 (MEG3) is abundantly expressed in many tissues, and plays a major role in growth and development [29]. MEG3 is also strongly expressed during palatogenesis; its inhibition by proliferation of palatal mesenchymal cells involved in the TGF-β/Smad pathway can result in cleft palate in mouse fetuses exposed to TCDD [30]. A study on the mechanism of TCDD toxicity found that 100% of fetuses exposed to a specific concentration of TCDD at 12.5 days of fetal life developed cleft palate; however, when the same concentration of TCDD was administered to knockout mice, no cleft palate developed, suggesting that the presence of aryl hydrocarbon receptor (AhR) is important for the development of cleft palate via TCDD exposure during the fetal period [31]. AhR is a ligand-dependent transcription factor that is activated when TCDD binds to cytoplasmic AhR in vivo. However, when TCDD was administered to AhR-deficient pregnant mice, no fetal malformations occurred, suggesting that AhR is essential for the teratogenicity of TCDD. In addition, AhR in the palate is mostly expressed in epithelial tissues and less expressed in mesenchymal tissues [32,33].

The mechanism of TCDD-induced cleft palate has been reported as follows: Cleft palate is caused by either failure of the palatine processes to make contact because of insufficient horizontal elongation of the palatine processes [31] or failure of the left and right palatine processes to fuse because of inability of the epithelial cord to disappear after contact with the palatine processes [34]. In this study, we investigated the mechanism of cleft palate induction using a stereomicroscope. Thirty-eight TCDD-treated mouse fetuses showed anterior palatal fusion and posterior palatal separation under a stereomicroscope at 15 days of age; among them, 15 also exhibited anterior palatal fusion and posterior separation in the histological analysis.

When the left and right palatine processes fuse after contact, epithelial cells lose their polarity in the epithelial cord and either reconstruct their cytoskeleton to convert to mesenchymal cell traits, which is called the epithelial–mesenchymal transition (EMT), or undergo apoptosis [35,36,37,38,39,40]. In this study, E-cadherin-, β-catenin-, and α-catenin-positive cells were found in the midline epithelial cord of the anterior palate in the TCDD-treated group, and TUNEL-positive cells were found in the same area. Thus, the palate was in the process of fusion. In addition, in the posterior palate, E-cadherin-, β-catenin-, and α-catenin-positive cells were found in the epithelial cords near the detachment, and TUNEL-positive cells were found around the detachment of the palate. In the control group, the left and right palatine processes were in the process of fusion, and TUNEL-positive cells resulting from the EMT were observed around the remaining epithelial cord in the posterior median palate. However, in the TCDD group, apoptosis was observed in the anterior mid-palate. In the anterior part, apoptosis was similar to that of the control group, and was considered to occur during palatal fusion. In the posterior part of the palate, the palate tissue was ruptured, and TUNEL-positive cells were found around the ruptured part of the palate, suggesting that the rupture occurred during palatal fusion.

Gao et al. [41] reported the possibility of cleft palate development in TCDD-treated mice through inhibited differentiation of cells in the epithelial cord via accelerated cell proliferation. In this study, Ki67-positive cells were found around the posterior palatal cleft in the TCDD-treated group, suggesting that abnormal cell proliferation may inhibit the differentiation of epithelial cord cells during palatal fusion, resulting in palatal cleft formation [42].

Cancer invasion and metastasis are phenomena in which adherent tissues are torn. When cancer cells migrate to other tissues, they are assumed to disrupt the basement membrane multiple times as the first step [43,44]. The basement membrane exists between epithelial and mesenchymal tissues and is composed of laminin, type IV collagen, heparan sulfate proteoglycans, entactin, and perlecan [45,46]. Laminin, in particular, is involved in cell adhesion and assumed to be involved in the metastatic potential of cells [47]. Moreover, laminin and collagen are extracellular matrices that fill the extracellular space and are responsible for cell adhesion, tissue development, and maintenance of the structure [48]. We have already reported the influence of the basement membrane on the mechanism of palatal cleft after palatal fusion [24].

TCDD may cause fragility in epithelial tissues [49,50,51]. This suggests that the epithelial tissue may be abnormal in the nasal mucosa after palatal fusion. As β-catenin is reportedly involved in the signaling pathway of epithelial cord loss via the EMT [52], the loss of β-catenin-positive cells in the nasolateral mucosal epithelium anterior to the palate in the TCDD-treated group may be due to an abnormality in the signaling pathway using β-catenin as a ligand, which prevents the normal loss of epithelial cords and prevents palatal fusion. Epithelial cells construct epithelial tissues via intercellular adhesion, and E-cadherin is an intercellular adhesion factor expressed on cell membranes. In cancer cells, E-cadherin binds to β-catenin in the cytoplasm, and β-catenin binds to α-catenin [53], forming a triadic complex on epithelial cell membranes. When E-cadherin expression is decreased in cancer cells, the expression of intercellular adhesion factors and the extracellular matrix is lost [54]. Moreover, E-cadherin is a representative epithelial marker [55]. The loss of E-cadherin in TCDD-treated palates suggests that the loss of basement membrane and epithelium in these palates may be due to a lack of epithelial tissue strength resulting from reduced adhesion of basement membrane and epithelial cells. This suggests that the mechanism of palate dehiscence may be similar to that of cancer metastasis.

According to previous reports and the results of this study, the mechanism of cleft palate development after fusion may be complicated by rupture of the basement membrane and abnormal proliferation of epithelial tissue in the once-fused palate, as well as reduced intercellular adhesion and the inhibition of mesenchymal cell proliferation, resulting in abnormal EMT and the development of cleft palate.

Previously, it was generally believed that cleft palate was caused by failure of the left and right palatal processes to fuse during palatogenesis. Although various studies have analyzed the causes of palatal fusion failure [6,7,8,9,31,34], they have not been able to prevent the disease. In contrast, this study showed that cleft palate may occur when the already-fused palate begins to separate, indicating that separation after fusion of the palate is one of the pathogenesis mechanisms of cleft palate. Thus, future research may be able to identify the factors that prevent separation of the palate, which could lead to a new method for preventing the development of cleft palate.

## 4. Materials and Methods

### 4.1. Experimental Animals and Embryo Extraction

Eight- to 10-week-old Institute of Cancer Research (ICR) female mice (CLEA, Tokyo, Japan) were purchased and maintained at an appropriate room temperature (22 ± 1 °C) and humidity (50 ± 10%) under regular 12-h light/dark cycles with unrestricted access to standard solid chow and drinking water. Female heifers were mated overnight with mature male mice of the same strain, and vaginal plug formation was confirmed the next morning, which was assumed to represent a pregnancy at midnight on the same day; this was designated embryonic day 1. On day 12 of gestation, 0.4 mL of olive oil containing toluene was administered to the control group, and 40 μg/kg of TCDD (Accu Standard Inc., New Haven, CT, USA), which is the concentration that results in cleft palate in 100% of mice, was dissolved in 0.4 mL of olive oil containing toluene and administered as a single dose by gastric tube (Accu Standard Inc., New Haven, CT, USA). To observe the palate during the palatogenesis period, fetuses removed by cesarean section were used in this study. Thirty-eight mouse fetuses were removed at 15 days of fetal age, which exhibited the highest frequency of palate fusion in previous studies [23], and 38 fetuses with anterior palate fusion and posterior palate separation were used in the TCDD group. As the stage of palatogenesis was delayed by approximately 1 day in the TCDD-treated group compared to that in the control group [56], four mouse fetuses at 14 days of age were used as the control group. The study protocol was approved by the Animal Care and Use Committee of the School of Dentistry, Aichi Gakuin University (approval No. AGUD404, 2018). Animal care and experimental procedures were conducted in accordance with the Regulation on Animal Experimentation at the School of Dentistry, Aichi Gakuin University.

### 4.2. Tissue Fixation and Section Preparation

The removed fetal heads were immersed in 4% paraformaldehyde in 0.1 M of phosphate buffer solution (PBS) with a pH of 7.4 for 24 h. The cells were dehydrated, degreased using an ascending ethanol series and xylene, and embedded in paraffin. Frontal sections of the fetal heads with a thickness of 6 μm were prepared using a rotary microtome.

### 4.3. Histological Observation

H-E staining was performed on the frontal sections of the fetal palate to observe the palate from the anterior to the posterior end during palatogenesis, and the results were compared between the control and TCDD-treated groups.

### 4.4. Immunohistological Observations

#### 4.4.1. Observation of Epithelial Tissue, Interepithelial Cell Adhesion Factor, and Basement Membrane Tissue via Fluorescent Immunohistochemistry

Mouse monoclonal anti-E-cadherin antibody (ab76055, Abcam, Cambridge, UK) and rabbit polyclonal anti-laminin antibody (L9393, Sigma, St. Louis, MO, USA) were used as primary antibodies. Anti-E-cadherin antibody was used to observe epithelial tissue and interepithelial cell adhesion factors, and anti-laminin antibody was used to observe the basement membrane tissue.

Fetal palatine frontal sections were deparaffinized and exposed to microwave irradiation in boiling citrate buffer (pH 7.0) for 5 min for antigen activation, then treated with 15,000 U/mL hyaluronidase (H0164, Tokyo Chemical Industry, Tokyo, Japan)/0.01 M PBS at 37 °C for 30 min. After blocking with 5% Block Ace (UKB80, DS Pharma Biomedical, Osaka) for 2 h, the cells were immersed in a mixture of primary antibodies diluted to a concentration optimized by preliminary experiments, shaken for 1 h at room temperature, and further incubated at 4 °C overnight. After 1 day, the secondary antibodies were reacted for 2 h at room temperature, washed, sealed, and observed from the anterior to posterior palate. The secondary antibodies used in this study were goat anti-mouse IgG (H + L), Alexa Fluor 488 (A-11029, Invitrogen, Carlsbad, CA, USA), goat anti-rabbit IgG (H + L), and Alexa Fluor 594 (R37117, Invitrogen, Waltham, MA, USA).

#### 4.4.2. Observation of Epithelial Cell Adhesion Factors

Rabbit monoclonal anti-β-catenin antibody (ab32572, Abcam) and mouse monoclonal anti-α-catenin antibody (66221-1-lg, Proteintech, Rosemont, IL, USA) were used as primary antibodies to observe interepithelial cell adhesion factors from the anterior to posterior palate.

Antigen activation was conducted via deparaffinization of the anterior section of the fetal palate and microwave irradiation in boiling citrate buffer (pH 7.0) for 5 min, followed by blocking with 5% Block Ace (UKB80, DS Pharma Biomedical, Osaka, Japan) for 2 h. The cells were then immersed in a mixture of primary antibodies diluted to a concentration optimized by preliminary experiments, shaken for 1 h at room temperature, and further incubated at 4 °C overnight. The next day, the cells were reacted with the labeled polymer corresponding to the primary antibody for 30 min at room temperature, then stained using a peroxidase staining kit (Nova RED Substrate Kit, SK-4800, Vector, Burlingame, CA, USA) for 5–10 min at room temperature.

#### 4.4.3. Observation of Apoptosis

Fetal palatine frontal sections were deparaffinized and subjected to a series of manipulations using the TUNEL assay kit (ab66110, Abcam) for fluorescent TUNEL staining. Nuclei were stained with DAPI (D9542; Sigma). Apoptosis was observed from the anterior to posterior palate.

#### 4.4.4. Observation of Cell Proliferation

The primary antibody used in this study was mouse monoclonal anti-Ki67 antibody (RM-9106, Lab Vision, Fremont, CA, USA), and cell proliferation was observed from the anterior to posterior palate. Antigen activation treatment was performed by deparaffinization of the anterior section of the fetal palate and microwave irradiation in boiling citrate buffer (pH 7.0) for 5 min. The cells were then immersed in a mixture of primary antibodies diluted to a concentration optimized by preliminary experiments, shaken for 1 h at room temperature, and further incubated at 4 °C overnight. The next day, the cells were reacted with the labeled polymer corresponding to the primary antibody for 30 min at room temperature, then stained using a peroxidase staining kit (Nova RED Substrate Kit, SK-4800, Vector) for 5–10 min at room temperature. All tissue sections subjected to H&E staining and immunohistochemistry were examined and photographed using an all-in-one fluorescence microscope (BZ-X710, Keyence, Osaka, Japan).

## 5. Conclusions

In fetal mice exposed to TCDD at concentrations that cause cleft palate in 100% of cases, some of the fetuses exhibited cleft palate dehiscence during palatal fusion. The mechanism of cleft palate generation by cleavage after palatal fusion in TCDD-treated mice may be similar to that of cancer metastasis. Cleft palate development after fusion may be caused by an abnormal EMT resulting from the complex effects of abnormal proliferation of epithelial tissues, abnormal intercellular adhesion, and inhibition of mesenchymal cell proliferation along with rupture of the basement membrane.

By elucidating the mechanism of cleft palate cleavage, this research may lead to the identification of factors that prevent cleft palate cleavage and the establishment of new preventive methods.

## Figures and Tables

**Figure 1 ijms-23-02069-f001:**
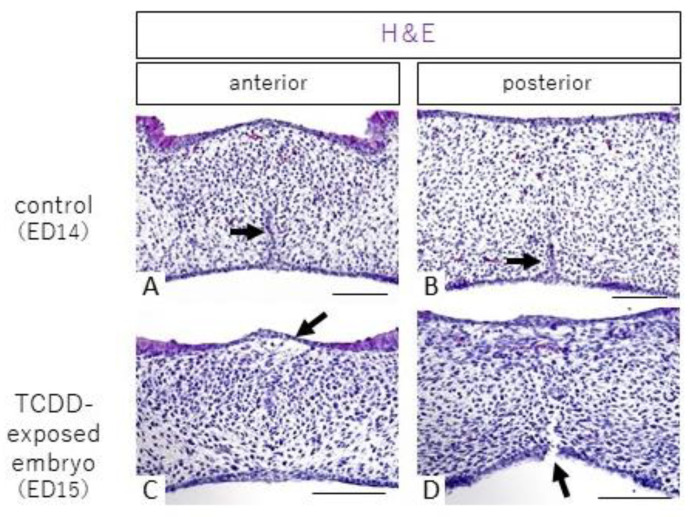
H&E staining images of palatal frontal sections from control and TCDD-treated groups. (scale bar: 100 µm). In the control group, the palate was fused from the anterior to posterior end at 14 days of age (**A**,**B**), and epithelial cords were present (**A**,**B**, arrow). The palatal shelf of the control group shows the presence of uniform cells throughout (**A**,**B**). In the TCDD group, there was fusion in the anterior part of the palate (**C**) and separation in the posterior part of the palate (**D**). In the anterior part of the palate of the TCDD-treated group, the submucosal cell density on the nasal side was sparse (**C**, arrow). The posterior part of the palate showed separation of the palate from the oral side (**D**, arrow).

**Figure 2 ijms-23-02069-f002:**
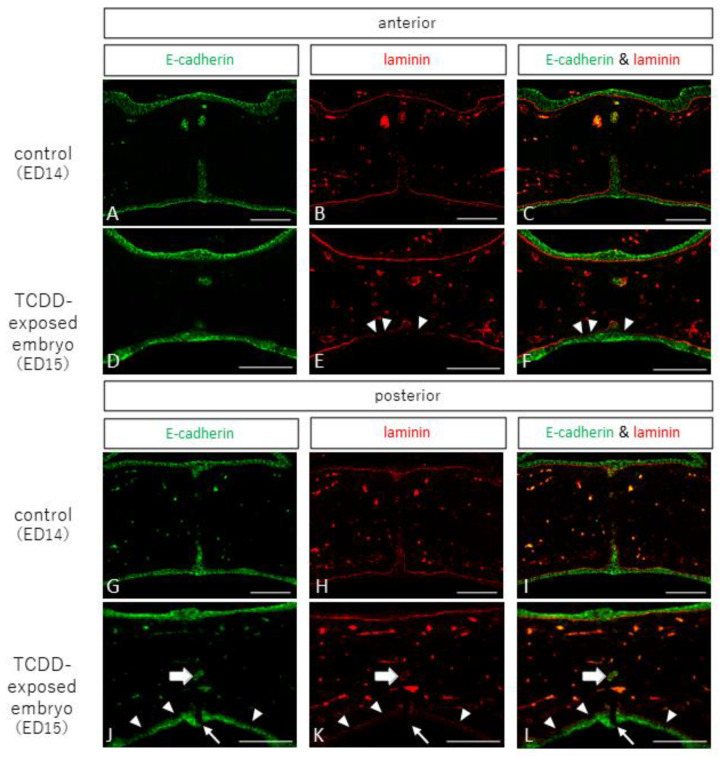
E-cadherin and laminin staining images of palatal frontal sections from control and TCDD-treated groups (scale bar: 100 µm). In the control group, E-cadherin was positive for epithelial cells in the nasal and oral mucosa and epithelial cords in the anterior and posterior palatal areas, and laminin showed continuous staining (**A**–**C**,**G**–**I**). In the TCDD group, E-cadherin was positive for epithelial cells in the nasal and oral mucosa and epithelial cords at the anterior palatal fusion site; laminin was stained continuously in the nasal mucosa and epithelial cords (**D**–**F**) but showed discontinuous staining in the oral mucosa (**E**,**F**, arrowheads). In the posterior part of the palate, there was a break in the mucosa on the oral side (**J**–**L**, small arrows). E-cadherin was positive for epithelial cells in the mucosa on the nasal and oral sides, and laminin was stained continuously in the mucosa on the nasal side (**J**–**L**). However, laminin was also stained discontinuously in the mucosa on the oral side, even outside the break (**K**,**L**, arrowheads). The epithelial cells in the same area were negative for E-cadherin (**J**–**L**, arrowheads). In addition, E-cadherin was positive in epithelial cells in the epithelial cord near the palatine detachment (**J**–**L**, large arrows), and laminin staining was observed (**K**,**L**, large arrows).

**Figure 3 ijms-23-02069-f003:**
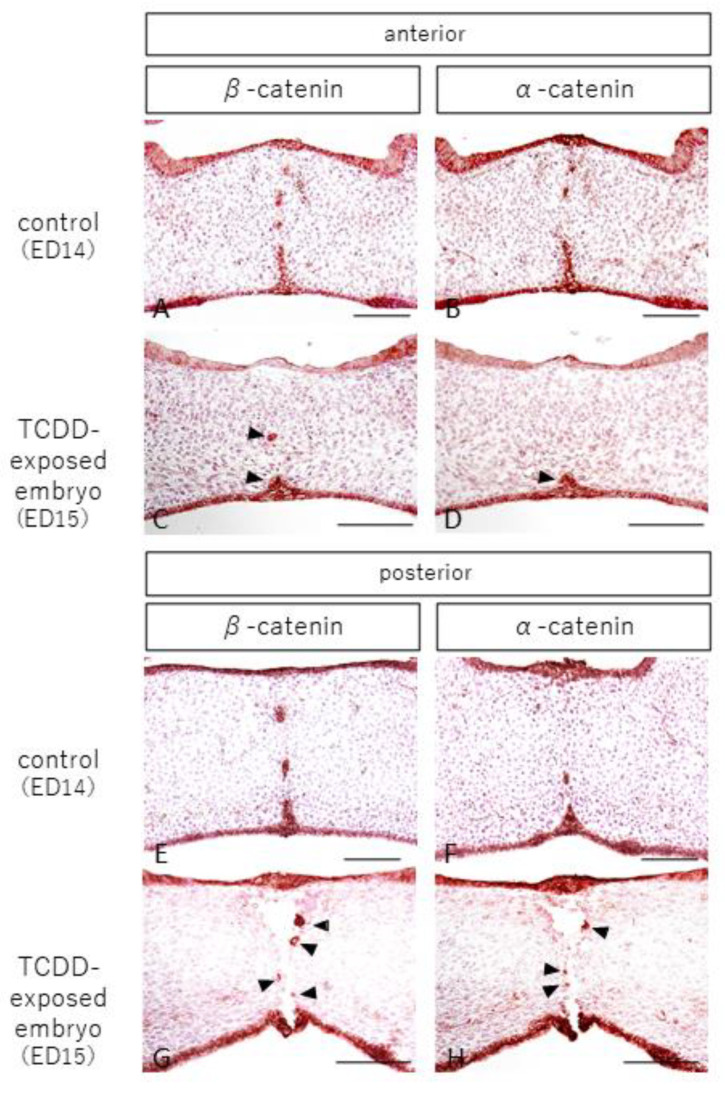
Staining images of β-catenin and α-catenin in the palatal frontal sections of control and TCDD-treated groups (scale bar: 100 µm). In the control group, β-catenin and α-catenin were positive for epithelial cells in the anterior and posterior palate, the nasal and oral mucosa, and the epithelial cord (**A**,**B**,**E**,**F**). In the TCDD group, β-catenin and α-catenin were positive for epithelial cells in the nasolacrimal and oral mucosa at the anterior palatal fusion but negative in the median nasolacrimal mucosa (**C**,**D**). β-catenin and α-catenin were positive for epithelial cells in the epithelial cord of the midline palate (**C**,**D**, arrowheads). In the posterior palatal detachment, β-catenin and α-catenin were positive for epithelial cells in the nasal and oral mucosa (**G**,**H**) and in the epithelial cord near the palatal detachment (**G**,**H**, arrowheads).

**Figure 4 ijms-23-02069-f004:**
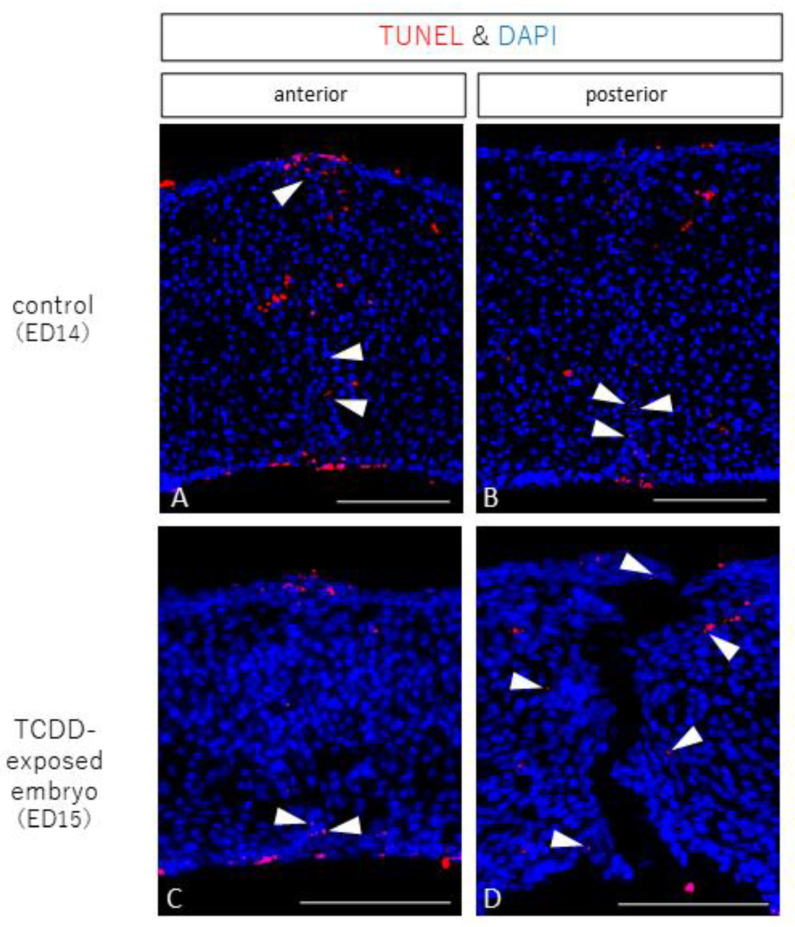
TUNEL stained images of palatal frontal sections from control and TCDD-treated groups (scale bar: 100 µm). In the control group, TUNEL-positive cells were found in the epithelial cords in the anterior and posterior palates (**A**,**B**, arrowheads). In the TCDD-treated group, TUNEL-positive cells were found in the epithelial cords in the anterior palate (**C**, arrowhead), as in the control group. In the posterior palate, TUNEL-positive cells were found around the dissection area (**D**, arrowhead).

**Figure 5 ijms-23-02069-f005:**
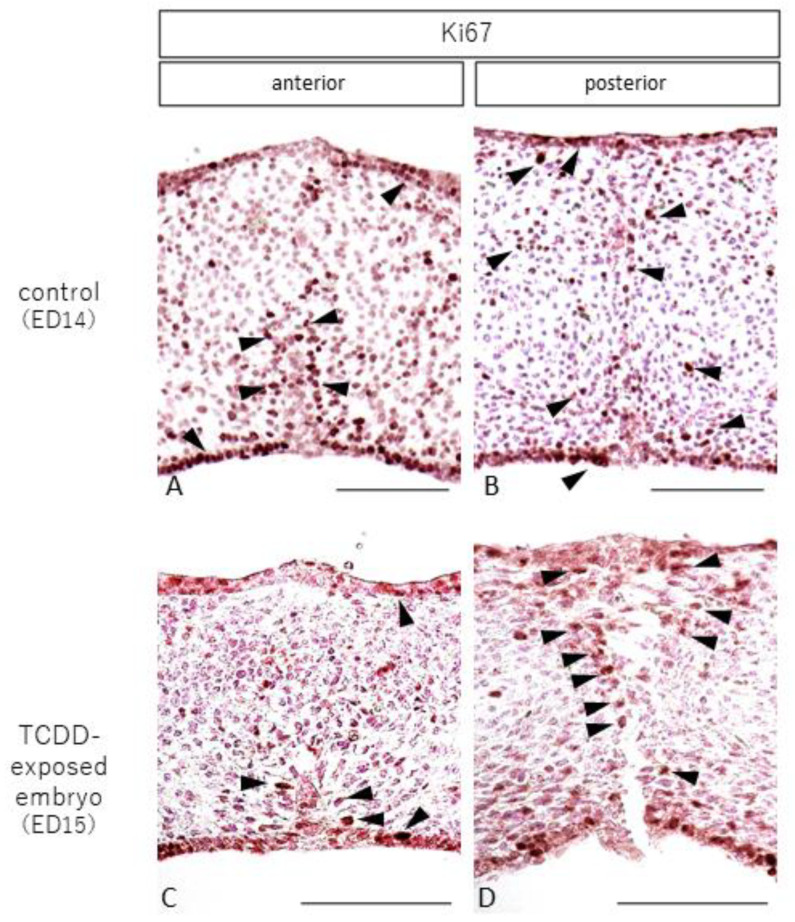
Ki67-stained images of anterior palatal sections from control and TCDD-treated groups (scale bar = 100 µm). In the control group, Ki67-positive cells were found in the anterior palate on the nasal and oral sides of the mucosa and around the epithelial cord (**A**, arrowhead). In the posterior palate, Ki67-positive cells were found in the nasal and oral mucosa and the entire palatal shelf (**B**, arrowhead). In the TCDD-treated group, Ki67-positive cells were found in the anterior palatal fusion area on the nasal and oral sides of the mucosa and around the epithelial cord (**C**, arrowhead). In the posterior part of the palate, Ki67-positive cells were found around the dissection area (**D**, arrowhead).

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
