# Peer review of "Histological and Immunohistochemical Studies to Determine the Mechanism of Cleft Palate Induction after Palatal Fusion in Mice Exposed to TCDD"

_ijms, 2022, doi:10.3390/ijms23042069_

Round 1

Reviewer 1 Report

The contents in this paper were well organized and the readability is very good.

Specially, the interpretation of the experimental results for the mechanism of cleft palate is excellent. However, there are some points that are difficult to understand. Please correct this section and provide additional explanation.

1. in line 208, add a description about AhR.

2. in line 226, TUNEL-positive cells are also found in the posterior palate(Figure 4). what is the difference from the control group? and what is the difference in apoptosis between two groups.

3. in lne 48, remove “:” form “the:”

Author Response

Response to reviewers

Please find our responses to each comment below. The reviewer comments are in italics and our responses are in normal font.

1) In line 208, add a description about AhR.

The following text has been added to lines 203–208:

AhR is a ligand-dependent transcription factor that is activated when TCDD binds to cytoplasmic AhR in vivo. However, when TCDD was administered to AhR-deficient pregnant mice, no fetal malformations occurred, suggesting that AhR is essential for the teratogenicity of TCDD [?]. In addition, AhR in the palate is mostly expressed in epithelial tissues and less expressed in mesenchymal tissues [36, 37].

2) In line 226, TUNEL-positive cells are also found in the posterior palate (Figure4). What is the difference from the control group? And what is the difference in apoptosis between two groups.

The following text has been added to lines 226–233:

In the control group, the left and right palatine processes were in the process of fusion, and TUNEL-positive cells resulting from the EMT were observed around the remaining epithelial cord in the posterior median palate. However, in the TCDD group, apoptosis was observed in the anterior mid palate. In the anterior part, apoptosis was similar to that of the control group, and was considered to occur during palatal fusion. In the posterior part of the palate, the palate tissue was ruptured, and TUNEL-positive cells were found around the ruptured part of the palate, suggesting that the rupture occurred during palatal fusion.

3) In line 48, remove “:”form ”the:”

 I removed “:” form “the: “ in line 51.

Reviewer 2 Report

Generally it is a well written manuscript. However the manuscript can be improved by addressing the following comments:

1) Cleft lip and palate are common craniofacial deformities. Deshpande et al 2018 provides overview of the multiple genes and molecular pathways that have been implicated in palatal fusion. Also, Paiva et al 2019 discuss the role of genes involved in ECM composition and remodeling during secondary palate formation and pathogenesis and genetic etiology of CL/P. Please include it in your discussion and cite:

  • Deshpande AS, Goudy SL. Cellular and molecular mechanisms of cleft palate development. Laryngoscope Investig Otolaryngol. 2018;4(1):160-164. Published 2018 Nov 15. doi:10.1002/lio2.214
  • Paiva KBS, Maas CS, Dos Santos PM, Granjeiro JM, Letra A. Extracellular Matrix Composition and Remodeling: Current Perspectives on Secondary Palate Formation, Cleft Lip/Palate, and Palatal Reconstruction. Front Cell Dev Biol. 2019 Dec 13;7:340. doi: 10.3389/fcell.2019.00340. PMID: 31921852; PMCID: PMC6923686.

2) Also, the current first author has already published the paper regarding the same topic in 2018 ”Sakuma C, Imura H, Yamada T, Sugahara T, Hirata A, Ikeda Y, Natsume N: Cleft palate formation after palatal fusion occurs 433 due to the rupture of epithelial basement membranes. J. Craniomaxillofac. Surg, 46(12):2027-2031,2018 “. Please explain the additive value of the currently published paper. What is novel in the current paper, not just repetition of the previously published paper.

Author Response

Response to the reviewer

Please find our responses to each comment below. The reviewer comments are in italics and our responses are in normal font.

1) Cleft lip and palate are common craniofacial deformities. Deshpande et al 2018 provides overview of the multiple genes and molecular pathways that have been implicated in platal fusion. Also, Pavia et al 2019 discuss the role of genes involved in ECM composition and remodeling during secondary palate formation and pathogenesis and genetic etiology of CL/P. Please include it in your discussion and cite:

Doi: 10.1002/lio2.214    Doi: 10.3389/fcell.2019.00340

The following text has been added to lines 251–256:

As β-catenin is reportedly involved in the signaling pathway of epithelial cord loss via the EMT [57], the loss of β-catenin-positive cells in the nasolateral mucosal epithelium anterior to the palate in the TCDD-treated group may be due to an abnormality in the signaling pathway using β-catenin as a ligand, which prevents the normal loss of epithelial cords and prevents palatal fusion.

The following text has been added to lines 251–256:

Moreover, genetic mutations and genetic polymorphisms in several genes, including extracellular matrix genes, soluble factors, and enzymes responsible for extracellular matrix remodeling, may play a role in the etiology of cleft palate [32]. In addition, maternally expressed gene 3 (MEG3) is abundantly expressed in many tissues, and plays a major role in growth and development [33]. MEG3 is also strongly expressed during palatogenesis; its inhibition by proliferation of palatal mesenchymal cells involved in the TGF-β/Smad pathway can result in cleft palate in mouse fetuses exposed to TCDD [34].

2) Also, the current first author has already published the paper regarding the same topic in 2018 “Sakuma C, Imura …”Please explain the additive value of the currently published paper. What is novel in the current paper, not just repetition of the previously published paper.

In our previously published paper, we reported the mechanism of palatal dehiscence after palatal fusion via rupture of the basement membrane. In this paper, as well as rupture of the basement membrane, we report for the first time that cleft palate separation after palatal fusion is also caused by the complex effects of abnormal proliferation of epithelial tissues, abnormal intercellular adhesion, and inhibition of mesenchymal cell proliferation, which result in an abnormal epithelial-mesenchymal transition and development of a cleft palate.

I have highlighted the contribution of this study in the Conclusions section.

Reviewer 3 Report

Dear Sirs, thank you for the opportunity to review this interesting paper. Unfortunately, there are some unacceptable omissions in your paper, that I would like to mention

  •  
  • In the abstract there should be stated in 1st sentence that the paper concerns mice, it shouldn't refer to previous studies
  • In introduction, please state the differences in racial occurrence of CLP (the frequency in Asian citizens is higher)
  • The materials and methods are the last part of the paper, so this should be highlightened earlier that the paper concerns mice
  • I find results and discussion the strongest part of this article
  • I would personally change the type of writing to 3rd person, meaning there should be "In the laboratory where the observations were made..." not "in our laboratory" - etc. Please correct that style
  • The paper is full of figures, which is very good at that point
  • The references have double numbers. Please, correct that
  • Please correct references due to standards (+ there are too many citations of one of the Authors of the manuscript)
  • the topic is quite novel and the referecnes should base on the last 10-15 years (among them there are a lot of older ones

Unfortunately, due to those suggestions, I need to give you a negative recommendation but I would like to highlight that after correcting it, you should upload the article once more.

Author Response

Response to the reviewer

Please find our responses to each comment below. The reviewer comments are in italics and our responses are in normal font.

1) In the abstract there should be stated in 1st sentence that the paper concerns mice, it shouldn't refer to previous studies.

I have added text to the abstract stating that the study was performed using mice. The reference to our previous study has been deleted; instead, our finding that the basement membrane of fused palate tissue can cause the palate to separate after fusion in mice has simply been stated as background to the study.

2) In introduction, please state the differences in racial occurrence of CLP (the frequency in Asian citizens is higher).

Lines 36–37 have been corrected as follows:

The incidence of cleft lip and cleft palate is highest among Asians, with an incidence of 1 in 500 to 600 in the Japanese population [1–3].

3) The materials and methods are the last part of the paper, so this should be highlightened earlier that the paper concerns mice.

This has been added to the introduction (Lines 77–78).

4) I find results and discussion the strongest part of this article. I would personally change the type of writing to 3rd person, meaning there should be "In the laboratory where the observations were made..." not "in our laboratory" - etc. Please correct that style.

All such references have been corrected throughout the article.

5) The paper is full of figures, which is very good at that point. The references have double numbers. Please, correct that.

The reference list has been corrected according to the journal formatting requirements.

6) Please correct references due to standards (+ there are too many citations of one of the Authors of the manuscript). The topic is quite novel and the references should base on the last 10-15 years (among them there are a lot of older ones).

As suggested, the manuscript has been updated to include more recent papers and works by other authors.

Round 2

Reviewer 3 Report

Dear Sirs, 

to my mind you did a good job by correcting the paper, but still for me the ratio of self-cites is too high. Please, try to replace some articles (I think that the total maximum would be 10 for that article and still I think it is too much). There you have some articles similar to yours. Please, try to replace them:

  1. Xu J, Liu F, Xiong Z, Huo J, Li W, Jiang B, Mao W, He B, Wang X, Li G. The cleft palate candidate gene BAG6 supports FoxO1 acetylation to promote FasL-mediated apoptosis during palate fusion. Exp Cell Res. 2020 Nov 15;396(2):112310. doi: 10.1016/j.yexcr.2020.112310
  2. Wang X, Li C, Zhu Z, Yuan L, Chan WY, Sha O. Extracellular Matrix Remodeling During Palate Development. Organogenesis. 2020 Apr 2;16(2):43-60. doi: 10.1080/15476278.2020.1735239
  3. Hutson MS, Leung MCK, Baker NC, Spencer RM, Knudsen TB. Computational Model of Secondary Palate Fusion and Disruption. Chem Res Toxicol. 2017 Apr 17;30(4):965-979. doi: 10.1021/acs.chemrestox.6b00350. 
  4. Jaruga, A.; Ksiazkiewicz, J.; Kuzniarz, K.; Tylzanowski, P. Orofacial Cleft and Mandibular Prognathism—Human Genetics and Animal Models. Int. J. Mol. Sci. 202223, 953. https://doi.org/10.3390/ijms23020953
  5. Vaivads, M.; Akota, I.; Pilmane, M. Cleft Candidate Genes and Their Products in Human Unilateral Cleft Lip Tissue. Diseases 20219, 26. https://doi.org/10.3390/diseases9020026

etc. Those are just examples, please, feel free to use other articles, but reduce self-citation. Thank you

Author Response

Response to reviewers

Please find our responses to each comment below. The reviewer comments are in italics and our responses are in normal font.

Reviewer 3

1) to my mind you did a good job by correcting the paper, but still for me the ratio of self-cites is too high. Please, try to replace some articles (I think that the total maximum would be 10 for that article and still I think it is too much). There you have some articles similar to yours. Please, try to replace them:

  1. Xu J, Liu F, Xiong Z, Huo J, Li W, Jiang B, Mao W, He B, Wang X, Li G. The cleft palate candidate gene BAG6 supports FoxO1 acetylation to promote FasL-mediated apoptosis during palate fusion. Exp Cell Res. 2020 Nov 15;396(2):112310. doi: 10.1016/j.yexcr.2020.112310
  2. Wang X, Li C, Zhu Z, Yuan L, Chan WY, Sha O. Extracellular Matrix Remodeling During Palate Development. Organogenesis. 2020 Apr 2;16(2):43-60. doi: 10.1080/15476278.2020.1735239
  3. Hutson MS, Leung MCK, Baker NC, Spencer RM, Knudsen TB. Computational Model of Secondary Palate Fusion and Disruption. Chem Res Toxicol. 2017 Apr 17;30(4):965-979. doi: 10.1021/acs.chemrestox.6b00350. 
  4. Jaruga, A.; Ksiazkiewicz, J.; Kuzniarz, K.; Tylzanowski, P. Orofacial Cleft and Mandibular Prognathism—Human Genetics and Animal Models. Int. J. Mol. Sci. 2022, 23, 953. https://doi.org/10.3390/ijms23020953
  5. Vaivads, M.; Akota, I.; Pilmane, M. Cleft Candidate Genes and Their Products in Human Unilateral Cleft Lip Tissue. Diseases 2021, 9, 26. https://doi.org/10.3390/diseases9020026

etc. Those are just examples, please, feel free to use other articles, but reduce self-citation. Thank you

As you suggested, we have reduced our own references to 10 and revised them to the new references.

For the paper before the revision, we removed references (2), (5), (6), (20), (23), (24), (25), (39), (47), and (51). In addition, references (4), (17), (19), (35), and (46) were added in this revision.

Round 3

Reviewer 3 Report

Thank you for the corrections